# Treatment of Infected Tibial Non-Unions with Ilizarov Technique: A Case Series

**DOI:** 10.3390/jcm9051352

**Published:** 2020-05-05

**Authors:** Gianluca Testa, Andrea Vescio, Domenico Costantino Aloj, Danilo Costa, Giacomo Papotto, Luca Gurrieri, Giuseppe Sessa, Vito Pavone

**Affiliations:** 1Department of General Surgery and Medical Surgical Specialties, Section of Orthopaedics and Traumatology, AOU Policlinico—Vittorio Emanuele, University of Catania, 95123 Catania, Italy; gianpavel@hotmail.com (G.T.); andreavescio88@gmail.com (A.V.); costa.danilo7@gmail.com (D.C.); giacomopapotto@gmail.com (G.P.); lca29@hotmail.it (L.G.); giusessa@unict.it (G.S.); 2Department of Traumatology, PO Sant’Andrea, 13100 Vercelli, Italy; aloj.domenico@gmail.com

**Keywords:** infected tibial non-union, Cierny–Mader classification, Ilizarov technique, bioactive glass, ASAMI scoring system

## Abstract

Background: The Ilizarov external fixation technique has been widely used for the treatment of long-bone infected non-unions. After surgical infected bone resection, to allow filling of the remaining bone gap, biomaterials with antibacterial properties could be used. The aim of this study was to report outcomes of infected tibial non-unions treated using the Ilizarov technique and antibacterial bioactive glass. Methods: Between April 2009 and December 2014, 26 patients with infected tibial non-unions were treated with the Ilizarov technique and possible use of the bioactive glass, S53P4. The Association for the Study and Application of Methods of Ilizarov (ASAMI) criteria, a clinical and radiographic evaluating tool, was used for assessing the sample. Results: The average age at the start of treatment was 51 years. The mean follow-up time was 113 weeks. According to the ASAMI Functional Scoring System, 10 excellent (38.5%) cases and 12 good (46.1%) values were recorded. According to the ASAMI Radiological System, they were excellent in 16 (61.5%) cases and good in nine (34.6%). Conclusions: Treatment of infected tibial non-unions using the Ilizarov technique was effective in bone segment regeneration. To fill the remaining bone gap, additional bioactive glass S53P4 could be used, allowing a decrease in re-interventions and minimizing complications.

## 1. Introduction

The Ilizarov external fixation technique has been used for the last five decades for the management of long-bone infected nonunions. This method uses percutaneously-inserted fine wires which are tensioned to provide a stiff and stable frame construct [1,2,3,4,5,6]. It permits the realization of compression, distraction, bone-lengthening, and deformity correction. It is a valid alternative treatment compared to internal fixation, especially when internal fixation can be complicated by bone loss, deformity, or failure of previous internal fixation [7]. Bone infection treatment after intramedullary nailing usually includes several surgical procedures, including removal of fixation devices, radical bone debridement with reaming of the canal, deep tissue sampling, elimination of infected bone, and/or insertion of local antibiotic delivery systems. In these cases, application of the Ilizarov external fixator is particularly useful even when there are significant soft tissue defects, which are often treated with musculocutaneous flaps. An advantage of the Ilizarov technique is stability of the fixation, allowing early weight-bearing ambulation and joint mobilization. After a wide bone resection using the elevator fragment technique, it is possible to carry out bone transport [8] through an osteotomy that is proximal or distal to the nonunion site.

The filling of an eventual bone gap is possible using bone substitutes that could be derived from biological products such as demineralized bone matrix, platelet-rich plasma, hydroxyapatite, or an adjunction of growth factors, or synthetic materials such as calcium sulfate, tri-calcium phosphate ceramics, bioactive glasses, or polymer-based substitutes [9]. Bioactive glasses are biocompatible, osteoconductive, and offer a porous structure with antibacterial properties which promotes their resorption and bone ingrowth [10]. Through a dissolution process that takes place as soon as the glass is implanted, the ion release at the bioactive glass surface induces an increase of pH and an osmotic pressure around the filling and kills both planktonic bacteria and bacteria in biofilm [10]. Although the topic is still not completely understood, recent findings report that bioactive glass S53P4 could be effective against multi-resistant strains without selection for resistance [10].

However, amputation is one of the most dangerous risks of long-bone infected nonunions and use of the Ilizarov technique can mean the difference between limb salvage and amputation. The aim of this study is to analyze the clinical and radiological outcomes of tibial infected non-unions treated with the Ilizarov technique and antibacterial bioactive glass.

## 2. Experimental Section

Between April 2009 and December 2014, 26 patients with an infected tibial nonunion were retrospectively analyzed. All patients were treated by a single surgeon and a multidisciplinary team of orthopedic and plastic surgeons, radiologists, infectologists, specialist nurses, and physiotherapists. According to the erythrocyte sedimentation rate and C-reactive protein, an empiric antibiotic therapy was administered. After intraoperative biopsy and bacterium identification, an appropriate antibiotic therapy was dispensed for two weeks to all patients, according to culture and sensitivity. Patients displaying negative culture were treated with four weeks of broad-spectrum antibiotics, covering Gram-positive and Gram-negative organisms.

Evaluated clinical and radiological parameters included gender, side of non-union affected, age, alcohol consumption, smoking and/or substance abuse, Cierny–Mader classification [11], open fractures Gustilo classification [12], previous surgical procedures, time from injury to Ilizarov frame, follow-up time, time to union, and post-treatment complications.

### 2.1. Surgical Technique

The Ilizarov technique was performed after the removal of previous fixation devices. These devices, together with bone and deep tissue samples taken from the nonunion site, were sent to the laboratory for culture. The tibial medullary canal was consistently reamed and debrided. A circular fixator was applied using transosseous wires and half-pins in order to avoid any additional soft tissue damage and minimize equinus ankle deformity. Then, an osteotomy of the tibial metaphysis was performed before beginning the bone transport. The residual bone gaps were filled with an implantation of 10 cc bioactive glass S53P4 granules (BonAlive^®^ BioMaterials LTD) with the capacity to simultaneously regenerate bone and prevent bacterial growth. If soft tissue coverage was required, this was done by the plastic surgeons during the same operation. Immediate post-operative weight-bearing with the aid of crutches was allowed. The antibiotic therapy was planned according to the antibiotic sensitivity that was based on histological and microbiological tests. Radiographic follow-ups were carried out after seven days and after one, three, six, and twelve months.

### 2.2. Clinical Assessment

All patients were clinically evaluated using the American Orthopaedic Foot and Ankle Score (AOFAS) [13] and the Association for the Study and Application of Methods of Ilizarov (ASAMI) criteria [14] at their last follow-up. The AOFAS incorporates subjective and objective criteria and is graded with a maximum score of 100 of which 50 are allocated to function, 40 to pain, and 10 to alignment. Following the functional ASAMI, the following outcomes were considered: excellent, if patients showed activity, no limping, minimal stiffness (loss of <15° knee extension/<15° ankle dorsiflexion), no reflex sympathetic dystrophy (RSD), and insignificant pain; good, if patients showed activity, with one or two of the other criteria; fair, if patients showed activity, with at least three other criteria; poor, if patients showed inactivity (unemployment or inability to perform daily activities because of injury); or failures in the event of amputation.

### 2.3. Radiographic Evaluation

Radiographic evaluation was conducted to reveal complications and to evaluate bone outcomes following application of the ASAMI criteria [14] at 6 months follow-up. In particular, bone ASAMI outcomes were considered: excellent, if there was union, no infection, deformity <7°, and limb length discrepancy <2.5 cm; good, if there was union and two of the other criteria; fair, if there was union and only one of the other criteria; or poor, if there was nonunion/refracture/union, infection, deformity > 7° and limb length inequality >2.5 cm.

### 2.4. Statistical Analysis

Continuous data are presented as means and standard deviations where appropriate. The analysis of variance (ANOVA) and the Tukey–Kramer method were used to compare AOFAS values. A chi-square test was used to compare ASAMI values. The selected threshold for statistical significance was *p* < 0.05.

## 3. Results

Of 26 patients, 20 (76.9%) were male and six (23.1%) were female. The left side was involved in 14 (53.8%) cases and the right in 12 (46.2%) cases. The mean age at start of treatment was 51.3 ± 15.3 (range 22–68) years.

The infected nonunion site was in the proximal segment in five tibias (19.2%), intermediate in 10 (38.5%), and distal in 11 (42.3%) (F _2,23_ = 1.18; *p* = 0.32). According to the Cierny–Mader classification [11], all patients had a diffuse infection (Type 4), of which 16 (61.5%) presented with host of type A and 10 (38.5%) with type B. The devices, removed before the Ilizarov external fixator was applied, included 11 plates (42.3%), seven intramedullary nails (26.9%), and eight external fixators (30.8%). Seven cases (26.9%) resulted from open fractures and were classified by Gustilo–Anderson [12] as 3A in four cases (15.4%) and as 3B in three cases (11.5%). The isolated microorganisms were staphylococci in 16 patients (61.5%) and multiple microorganisms in 10 (38.5%). The mean time between injury and Ilizarov external fixator application was 26 ± 13.7 weeks (range 9–65). The external fixation time was 9.8 ± 2.3 months (range 6–13). In seven patients (26.9%), the S53P4 bioactive glass was used for filling the residual bone gap.

The mean follow-up time was 113 ± 38.8 weeks (range 36–171). The clinical and radiographic bone results according to the ASAMI Scoring System were excellent in 10 cases (38.5%), good in 12 (46.1%), and fair in four cases (15.4%) (Table 1), while the functional results of the ASAMI Scoring System yielded excellent in 16 cases (61.5%), good in nine (34.6%), and fair in one (3.9%) (Table 2).

According to the ASAMI Functional Scoring System, no statistically significant differences were found between the different infected nonunion sites (*p* = 0.50) (Table 3), in terms of either the Cierny–Mader classification of chronic osteomyelitis (*p* = 0.28) (Table 4) or the usage of bioactive glass S53P4 (*p* = 0.99) (Table 5).

The overall average AOFAS score was 85.5 ± 6.2 (range 70–95): the average pain score was 37.5 ± 2.5 (range 31–40), the average function score was 38.4 ± 3.8 (31–45), and the average alignment score was 9.6 ± 0.6 (range 8–10) as shown in Table 6.

No statistically significant differences were found between the infected nonunion sites (Table 3) with regards to the Cierny–Mader classification of chronic osteomyelitis (Table 4) or the usage of bioactive glass S53P4 (Table 5) based on the AOFAS Scoring System.

Complications presented as external fixator pin infections in three cases (11.5%), Kirschner wire break in one (3.9%), and screw breakages in two cases (7.7%).

## 4. Discussion

This report found that the patients with infected tibial nonunions who were treated with the Ilizarov external fixation technique had excellent or good functional outcomes, regardless of the Cierny–Mader classification or fracture site. Moreover, only 15.4% of the sample reported deformities of <7° and limb-length discrepancy of <2.5 cm in the absence of infection. Of the 16 patients who were studied, six reported complications. In approximately one-third of the subjects the bone gap was filled with bioactive glass and 100% achieved excellent or good radiographic results.

An infected tibial nonunion is not only an orthopedic problem—it requires a multidisciplinary diagnostic and therapeutic approach. The infected tibial nonunion may be diagnosed when there are no signs of consolidation after six to eight months from the time of fracture and evident signs of local infection [15]. In fact, it is essential to obtain several clinical, laboratory, and imaging studies with the help of orthopaedics, physicians, infectologists, and microbiologists. With respect to laboratory tests, erythrocyte sedimentation rate and C-reactive protein values are usually higher during bone infections; however, these are non-specific tests that are not related to infection severity. Imaging such as plain X-ray, computed tomography, magnetic resonance imaging, and bone scans are also non-specific and are often used as support for diagnosis or surgery management. The gold-standard is a bone biopsy, which allows histological and microbiological tests to be done. However, a specific diagnosis may be done for an open lesion with purulent secretion after communicating with bone specialists or after surgery and germ isolation [16].

In order to manage the surgery after the diagnosis has been made, evaluation of infection extension is paramount. In our study, we referred to the Cierny–Mader classification in which the extension of bone infection and patients’ systemic responses were assessed. A diffuse infection, classified as Cierny–Mader type 4, occurred in all patients of which 40% presented with host type B, indicating they had partial immunodeficiencies and less positive prognoses. The treatment is both medical (with systemic antibiotics) and surgical (with debridement of necrotic bone and tibial stabilization). Complications could be malunion, limb-length discrepancy, joint rigidity, disuse osteoporosis, neurovascular damages, and soft tissue atrophy [17].

In this study we used the Ilizarov technique, which has been extensively used in the UK for over 15 years. It was designed by Gavriil Ilizarov in Kurgan, Russia in the 1950s but it became famous only after an Olympic jumper’s osteomyelitis was treated using this technique [1]. It consists of a circular external fixation system that allows for the correction of axis and rotatory deviation and the cover of soft tissue after bone resection. This technique is very useful, especially after significant loss of bone substance such as that seen in infected tibial nonunion debridement [18]. This surgical technique is based on the principle that long-bone resection, through the traction of the two bone fragments, stimulates the regeneration of new bone, called “regenerated bone”. The new bone originates from the periosteum or the endosteal shaft circulation [19,20]. It is preferable to perform a tibial osteotomy in the proximal or distal metaphysis since they are more vascularized [21].

Finally, the benefits of this method are different to those of other, similar techniques. The surgical procedure is less invasive, enables the adjustment of asymmetries and deformities, and allows immediate weight-bearing. The last point is important for the prevention of thromboembolic events, especially in elderly people [22]. The efficacy of the Ilizarov technique was demonstrated in several literature studies [7,23,24,25,26,27] as they reported optimal consolidation, good pain control, and quick recovery of daily activities in about 90% of the people who underwent the procedure (Table 7 and Table 8). Only one patient failed to recover daily activity and had pain and ankle stiffness.

In “regenerated bone” implants, Sadek et al. [28] noted no infection of the site and hypothesized that it could be a protective factor in secondary infection prevention. In our study, we did not find any septic relapse. The complications described in literature are the result of low patient tolerability due to the long-term use of the external fixator, pin-tract infections, pin or screw breakages, frequent edema, and joint limitations [29]. In 20% of people in this study, we found minor complications represented by pin-tract infections and/or wire and/or screw breakages. These complications could easily be solved since wires and screws can be cleaned or replaced.

Bone gaps resulting from surgery may need to be filled by surrogate bone or antibiotic-spacers [30]. The ideal material to replace bone tissue should be biocompatible, bioresorbable, osteoconductive, osteoinductive, porous, mechanically resistant, easy to use, safe, and cost-effective [31]. Autologous bone grafts represent the ideal choice for bone-filling because of their mechanical and biological features and low risk of immunogenicity or rejection [32]. However, autologous bone grafts require an additional surgical site (donor) and are associated with high cost and complications including chronic pain, dysesthesia, and infection [33].

Bone substitutes represent a good alternative, deriving from either biological or synthetic products. Among biological products, hydroxyapatite (HA) and β-tri-calcium phosphate represent good alternative bone substitutes [34,35,36]. The association of the two components, utilized in small bone defects, serves to inhibit the inflammatory response [34,35] and, at the same time, enhances osteoblast differentiation and accelerates osteogenesis [37,38]. Although it is associated with low rates of complications like infection or non-union [39], its usage is limited because of a mechanical resistance inferior to cancellous bone or bone allograft [40].

In 30% of our patients, to fill the bones gaps, it was necessary to use S53P4 bioactive glass granules. These granules are capable of stimulating osteoblast proliferation and differentiation and inhibiting bacterial proliferation [41], as an increase in pH and osmotic pressure and a decrease in antibiotic resistance were found [10]. These bioactive glass granules have been successfully used for more than 10 years after bone removal for tumors or in fracture treatment. They are reportedly effective in healing septic bone conditions [30,42], decreasing the hospitalization and wound complication rates in patients compared to other synthetic biomaterials that were used as carriers for antibiotics [29]. Bioactive glass granules have a resistance time comparable to autogenous bone [10,41]. Their use does not induce an inflammatory response and resorption is complete in six months for silica-based bioglasses [43]. More recently, phosphate- or borate-based bioglasses have been developed: these compositions reported a fast degradation leading to a possible match with the bone regeneration rate [44]. The side effect most commonly described in the literature is a lack of serous liquid in 2.5% of cases, especially in non-healed wounds [30]. Large bone gaps represent a relative contraindication: the outcome after BAG-S53P4 treatment might be related to proper filling of the cavities [10]. In our study, no significant differences were found between bioactive glass usage and non-usage according to AOFAS and ASAMI scores.

Limitations of the study included the small group of patients and follow-up, the retrospective nature of the study, and the lack of a control or comparative group of patients treated with other currently available surgical options or bone substitutes.

## 5. Conclusions

In conclusion, the Ilizarov technique is capable of promoting bone tissue generation. It is an easy procedure that allows immediate weight-bearing and a short hospitalization period. Use of the bioactive glass, S53P4, is ideal for filling bone gaps because it constitutes an excellent bone surrogate that has antibacterial and osteoinductive features. Despite no significant differences between bioactive glass addiction and no bioactive glass addiction, it should be used consistently in bone gaps in order to avoid other surgical interventions and, consequently, to reduce costs. However, this kind of surgery needs to be performed by an expert team and monitored using frequent post-operative follow-ups.

## Figures and Tables

**Table 1 jcm-09-01352-t001:** Bone results according to ASAMI Scoring System.

	Bone Results	Patients
Excellent	Union, no infection, deformity <7°, limb-length discrepancy <2.5 cm	10 (38.5%)
Good	Union + any two of the following: absence of infection, deformity <7°, limb-length discrepancy of <2.5 cm	12 (46.1%)
Fair	Union + only one of the following: absence of infection, deformity <7°, limb-length discrepancy of <2.5 cm	4 (15.4%)
Poor	Non-union/re-fracture/union + infection + deformity >7° + limb-length discrepancy >2.5 cm	0

**Table 2 jcm-09-01352-t002:** Functional results according to ASAMI Scoring System.

	Functional Results	Patients
Excellent	Active, no limp, minimum stiffness (loss of <15° knee extension/<15° ankle dorsiflexion), no reflex sympathetic dystrophy, insignificant pain	16 (61.5%)
Good	Active with one or two of the following: Limp, stiffness, reflex sympathetic dystrophy, significant pain	9 (34.6%)
Fair	Active with at least three of the following: Limp, stiffness, reflex sympathetic dystrophy, significant pain	1 (3.9%)
Poor	Inactive (unemployment or inability to return to daily activities because of injury)	0
Failures	Amputation	0

**Table 3 jcm-09-01352-t003:** Comparison between the results of the different infected nonunion sites using the AOFAS and ASAMI Bone and Functional Scoring System.

Infected Nonunion Site	AOFAS	ASAMI Bone					ASAMI Functional				
		E	G	F	P	*p* Value	E	G	F	P	*p* Value
Proximal	84.9 ± 5.8	4	3	2	0	*p* = 0.78	5	3	0	0	*p* = 0.50
Intermediate	86.8 ± 6.1	4	4	1	0		7	2	0	0	
Distal	84.8 ± 6.4	2	5	1	0		4	4	1	0	
Pro vs. Int	*p* = 0.63										
Int vs. Dis	*p* = 0.30										
Pro vs. Dis	*p* = 0.74										

**Table 4 jcm-09-01352-t004:** Comparison between the results of the different Cierny–Mader classification types using the AOFAS and ASAMI Bone and Functional Scoring System.

Cierny-Mader Classification	AOFAS	ASAMI Bone				ASAMI Functional		
		**E**	**G**	**F**	**P**	**E**	**G**	**F**
A	84.7 ± 6.0	6	7	3	0	8	7	1
B	86.8 ± 6.1	4	5	1	0	8	2	0
*p* value	*p* = 0.40	*p* = 0.83				*p* = 0.28		

**Table 5 jcm-09-01352-t005:** Comparison between the results of bioactive glass usage using the AOFAS and ASAMI Bone and Functional Scoring System.

Bioactive Glass Usage	AOFAS	ASAMI Bone				ASAMI Functional		
		**E**	**G**	**F**	**P**	**E**	**G**	**F**
No	85.8 ± 5.4	12	6	1	0	7	9	3
Yes	85.1 ± 7.1	4	3	0	0	3	3	1
*p* value	*p* = 0.79	*p* = 0.90				*p* = 0.99		

**Table 6 jcm-09-01352-t006:** Results according to AOFAS.

Criteria	Average Score (Range)
Pain	37.5 ± 2.5 (31–40)
Function	38.4 ± 3.8 (31–45)
Alignment	9.6 ± 0.6 (8–10)
Overall AOFAS score	85.5 ± 6.2 (70–95)

**Table 7 jcm-09-01352-t007:** Comparison between the results of the different studies using the ASAMI Bone Scoring System.

Authors	Patients	Frame	Excellent (%)	Good (%)	Fair (%)	Poor (%)
Present study	26	Ilizarov	38.5	46.1	3.9	0
Rohilla et al. [22]	35	Ilizarov	60	34.3	0	5.7
Maini et al. [23]	30	Ilizarov	70	10	0	20
Chaddha et al. [24]	25	Ilizarov	52	4	0	44
Yin et al. [25]	66	Ilizarov	67	23	7	3
Patil et al. [7]	78	Ilizarov	41	34	10	15
Farmanullah et al. [26]	58	Ilizarov	57	21	14	8

**Table 8 jcm-09-01352-t008:** Comparison between the results of the different studies using the ASAMI Functional Scoring System.

Authors	Patients	Frame	Excellent (%)	Good (%)	Fair (%)	Poor (%)	Failures
Present study	26	Ilizarov	61.5	34.6	3.9	0	0
Rohilla et al. [22]	35	Ilizarov	45.7	48.5	2.9	0	2.9
Maini et al. [23]	30	Ilizarov	27	40	10	23	0
Chaddha et al. [24]	25	Ilizarov	24	36	16	36	0
Yin et al. [25]	66	Ilizarov	40	43	17	0	0
Patil et al. [7]	78	Ilizarov	41	41	6	6	6
Farmanullah et al. [26]	58	Ilizarov	57	31	7	5	0

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
