# Peer review of "Treatment of Infected Tibial Non-Unions with Ilizarov Technique: A Case Series"

_jcm, 2020, doi:10.3390/jcm9051352_

Round 1
Reviewer 1 Report
In this study, the authors reported case series of treatment of infected nonunions with the Ilizarov technique. The authors followed total 26 patients who got treatment with the Ilizarov technique and S53P4 bioactive glass granules, and found by ASAMI scoring system that the treatment was effective in bone segment regeneration. This manuscript is well written, and would be useful information for treatment of infected nonunion.
I made couple of minor points.
1) Line 54-55, there may be a missing word between “the” and “is”. The sentence in the manuscript is "The aim of the is to analyze the clinical...", so it clearly lacks a word between "the" and "is". 2) Line 191-192, this sentence is unclear. It should be rewriting. This sentence "The authors who used the Ilizarov technique maintained that the occurrence of a “regenerated bone“ secondary infection was not possible (28)" is not easy to understood for me. Who is "the authors"? and what does it mean "The authors ... was not possible"?
Author Response
In this study, the authors reported case series of treatment of infected nonunions with the Ilizarov technique. The authors followed total 26 patients who got treatment with the Ilizarov technique and S53P4 bioactive glass granules, and found by ASAMI scoring system that the treatment was effective in bone segment regeneration. This manuscript is well written, and would be useful information for treatment of infected nonunion.
I made couple of minor points.
Q1) Line 54-55, there may be a missing word between “the” and “is”. The sentence in the manuscript is "The aim of the is to analyze the clinical...", so it clearly lacks a word between "the" and "is".
A1) Thanks for your comment. The mistake has been correct.
Q2) Line 191-192, this sentence is unclear. It should be rewriting. This sentence "The authors who used the Ilizarov technique maintained that the occurrence of a “regenerated bone“ secondary infection was not possible (28)" is not easy to understood for me. Who is "the authors"? and what does it mean "The authors ... was not possible"?
A2) Thanks for your comment. The phrase has been re-written.
Reviewer 2 Report
This is a retrospective single-center observational study assessing the outcome of infected tibial non-unions with the Ilizarov technique. The authors present the results of 26 patients, treated during a 5.75-year period. The minimal follow-up (36 months) was long enough to judge the results not only in terms of function, but also in terms of recovery of infection provided that the patients were not under suppressive antibiotics. The results are compared to published series with a similar or higher number of patients. The present series is well described regarding surgical technique and outcome. However, detailed information on the implant-associated infection is missing (duration of infection before Ilizarov therapy? Antimicrobial therapy (drug, biofilm-active drug, application modus, duration) before and after performing the Ilizarov technique? Follow-up after stopping antibiotics? Suppressive antibiotic therapy?) Information about these questions would improve the value of the manuscript for Infectious Disease specialists who participate in the multidisciplinary management of patients with implant-associated bone infections.
Specific comments
- Introduction, line 29. The statement “…used for the last several decades...” could be more precise, i.e. changed to “…used for the last 5 decades…”
- Introduction, line 51-2. The statement “…S53P4 has been shown to have a high potential against multi-resistant strains without selection for resistance” sounds like marketing for the product. To my knowledge, comparative randomized clinical studies are missing. Ref. 10 (Lindfors et al. 2017) is not a comparative study. Thus, it remains unclear whether control of infection was due to the bioactive glass or to antibiotics or to both.
- Experimental section, line 60. According to the protocol, all patients were managed with a multidisciplinary team. However, information about the Infectious Disease management (see above) is completely missing.
- Results, line 106. The statement “…two consumed alcohol” is not precise enough, if alcohol consumption is mentioned as risk factor. This information should either be modified by giving a definition of “alcohol consumption” or omitted, if quantitative information about alcohol consumption is not available in the charts.
- As mentioned above, information about the duration of infection, antibiotic therapy before Ilizarov management, type and duration of antimicrobial therapy after performing Ilizarov technique, and duration of antibiotic therapy after removing the Ilizarov device is completely missing. Pin tract infection and its management should be mentioned.
- Discussion, line 157. The term “x-ray” should be more precise, it probably means “plain x-ray”
Author Response
Reviewer 2:
This is a retrospective single-center observational study assessing the outcome of infected tibial non-unions with the Ilizarov technique. The authors present the results of 26 patients, treated during a 5.75-year period. The minimal follow-up (36 months) was long enough to judge the results not only in terms of function, but also in terms of recovery of infection provided that the patients were not under suppressive antibiotics. The results are compared to published series with a similar or higher number of patients. The present series is well described regarding surgical technique and outcome. However, detailed information on the implant-associated infection is missing (duration of infection before Ilizarov therapy? Antimicrobial therapy (drug, biofilm-active drug, application modus, duration) before and after performing the Ilizarov technique? Follow-up after stopping antibiotics? Suppressive antibiotic therapy?) Information about these questions would improve the value of the manuscript for Infectious Disease specialists who participate in the multidisciplinary management of patients with implant-associated bone infections.
Specific comments
Q1) Introduction, line 29. The statement “…used for the last several decades...” could be more precise, i.e. changed to “…used for the last 5 decades…”
A1) Thanks for your comment. The sentence has been correct.
Q2) Introduction, line 51-2. The statement “…S53P4 has been shown to have a high potential against multi-resistant strains without selection for resistance” sounds like marketing for the product. To my knowledge, comparative randomized clinical studies are missing. Ref. 10 (Lindfors et al. 2017) is not a comparative study. Thus, it remains unclear whether control of infection was due to the bioactive glass or to antibiotics or to both.
A2) Thanks for your comment. The phrase has been re-written.
Q3) Experimental section, line 60. According to the protocol, all patients were managed with a multidisciplinary team. However, information about the Infectious Disease management (see above) is completely missing.
A3) Thanks for your comment. The requested information was added.
Q4) Results, line 106. The statement “…two consumed alcohol” is not precise enough, if alcohol consumption is mentioned as risk factor. This information should either be modified by giving a definition of “alcohol consumption” or omitted, if quantitative information about alcohol consumption is not available in the charts.
A4) Thanks for your comment. The phrase has been removed.
Q5) As mentioned above, information about the duration of infection, antibiotic therapy before Ilizarov management, type and duration of antimicrobial therapy after performing Ilizarov technique, and duration of antibiotic therapy after removing the Ilizarov device is completely missing. Pin tract infection and its management should be mentioned.
A5) Thanks for your comment. The requested information was added.
Q6) Discussion, line 157. The term “x-ray” should be more precise, it probably means “plain x-ray”
A6) Thanks for your comment. The phrase has been correct.
Round 2
Reviewer 2 Report
The reviewers's criticisms have been adequately considered.